# Influence of Body Mass Index on Long-Term Outcome in Patients with Rectal Cancer—A Single Centre Experience

**DOI:** 10.3390/cancers11050609

**Published:** 2019-04-30

**Authors:** Maximilian Kalb, Melanie C. Langheinrich, Susanne Merkel, Christian Krautz, Maximilian Brunner, Alan Bénard, Klaus Weber, Christian Pilarsky, Robert Grützmann, Georg F. Weber

**Affiliations:** Department of Surgery, Erlangen University Hospital, Krankenhausstraße 12, 91054 Erlangen, Germany; max.kalb@gmx.net (M.K.); melanie.langheinrich@uk-erlangen.de (M.C.L.); susanne.merkel@uk-erlangen.de (S.M.); christian.krautz@uk-erlangen.de (C.K.); maximilian.brunner@uk-erlangen.de (M.B.); alan.benard@uk-erlangen.de (A.B.); klaus.weber@uk-erlangen.de (K.W.); christian.pilarsky@uk-erlangen.de (C.P.); robert.gruetzmann@uk-erlangen.de (R.G.)

**Keywords:** body mass index, rectal cancer, overall survival, distant metastasis

## Abstract

Background: Excess bodyweight is known to influence the risk of colorectal cancer; however, little evidence exists for the influence of the body mass index (BMI) on the long-term outcome of patients with rectal cancer. Methods: We assessed the impact of the BMI on the risk of local recurrence, distant metastasis and overall—survival in 612 patients between 2003 and 2010 after rectal cancer diagnosis and treatment at the University Hospital Erlangen. A Cox-regression model was used to estimate the hazard ratio and multivariate risk of mortality and distant-metastasis. Median follow up-time was 58 months. Results: Patients with obesity class II or higher (BMI ≥ 35 kg/m^2^, *n* = 25) and patients with underweight (BMI < 18.5 kg/m^2^, *n* = 5) had reduced overall survival (hazard ratio (HR) = 1.6; 95% confidence interval (CI) 0.9–2.7) as well as higher rates of distant metastases (hazard ratio HR = 1.7; 95% CI 0.9–3.3) as compared to patients with normal bodyweight (18.5 ≤ BMI < 25 kg/m^2^, *n* = 209), overweight (25 ≤ BMI <30 kg/m^2^, *n* = 257) or obesity class I (30 ≤ BMI <35 kg/m^2^, *n* = 102). There were no significant differences for local recurrence. Conclusions: Underweight and excess bodyweight are associated with lower overall survival and higher rates of distant metastasis in patients with rectal cancer.

## 1. Introduction

In 2014, more than 18,500 people in Germany were initially diagnosed with rectal cancer (11,414 males and 7281 females) [1], along with 7605 rectal cancer deaths (4519 in males and 3086 in females) [1]. The 5-year survival rate for rectal cancer increased in the 21st century from 44.3% in the early nineties (1990 to 1992) to 53.7% in the period of 2000–2002, earlier detection and progress in therapy being the cause [2]. Colorectal cancer is responsible for the third highest economic cost (13.1 Billion €, 10% of all cancer costs) in the European Union behind lung (18.8 billion €, 15%) and breast cancer (15.0 billion €, 12%). Regarding health care costs colorectal cancer (5.57 billion €, 11% of all-cancer related health care costs) was ranked second behind breast cancer (6.73 billion €, 13%) followed by prostate cancer (5.43 billion €, 11%) [3].

Obesity is a well-known risk-factor for the development of colorectal cancer [4,5]. Adiposity triggers several systemic and metabolic alterations that can influence carcinogenesis. It has been shown that obesity is related to enhanced levels of leptin in humans, leading to increasing total numbers of cells in different colonic tissues [6]. In addition, leptin was associated with reducing apoptosis in various in vitro experiments [6]. These observations suggest that, in some cases, an un-physiological elevation of leptin might become a risk factor for tumor growth.

In a prospective cohort study, performed in the period from 1986 until 1992, the Body Mass Index (BMI) of 47,723 male patients was correlated with the incidence of colorectal cancer or adenoma. Thereby, BMI was identified as a direct risk factor for the development of colorectal cancer independent of physical activity [7]. A meta-analysis of 31 studies performed in 2007 identified a dose-response correlation between BMI and colorectal cancer, showing that an increase of 2 kg/m^2^ elevated the risk for colorectal cancer by 7% [8]. Underweight impaired the early and long-term survival after rectal cancer surgery in a multicenter observational study [9].

However, the effect of the BMI on the long-term survival for patients with rectal carcinoma has not been assessed in detail. The aim of our study was therefore to investigate the influence of BMI on the long-term survival of patients diagnosed with rectal cancer.

## 2. Results

### 2.1. Subject Characteristics

The median follow-up time for 612 patients was 93 (range 2–156) months. The male-to-female ratio was 1.97 to 1. The mean age was 65 years (range 18–91), there was no significant difference for age <65 and ≥65 between the BMI-groups as classified by the World Health Organisation (WHO) or Group 1 and 2. Female patients were more likely to be underweight than male patients, whereas male patients showed higher rates of overweight, obesity class I and obesity class II. Underweight and obese patients had a higher perioperative risk as classified by the American Society of Anaesthesiologists (ASA) scoring (1–4) (*p* = 0.005). An ASA-Score of 3 or 4 was found more often in individuals graded as underweight, obese class I, obese class II and obese class III. Group 2 showed a significantly higher rate of ASA-Score 3 or 4, compared to Group 1 (*p* = 0.004). CEA-levels were significantly more often elevated in Group 2 compared to patients in Group 1 (*p* = 0.048). Furthermore, patients in Group 2 were more likely to receive a multivisceral resection than patients in Group 1 (*p* = 0.042). Except for this, no significant differences for the tumour or patient characteristics could be demonstrated for Group 1 and 2. Further characteristics of the patients, tumours and the treatment details are shown in Table 1 and Table 2.

### 2.2. Quality Indicators

The median number of examined lymph nodes was 21 (range 2–76). Microscopic curative tumor resection rates were significantly lower in patients who were underweight (*p* = 0.031), without significant differences between patients in Group 1 and 2 (*p* = 0.238). Other than that, there was no discernible difference concerning the surgical quality indicators, showing no significance for aboral (*p* = 0.474) and circumferential (*p* = 0.445) resection margin or intraoperative local tumor cell dissemination (*p* = 0.186). The specimen quality was considered to be good in 555 patients (97.2%) without a significant dependency on BMI (WHO, *p* = 0.309; Group1 and 2, *p* = 0.203). The number of abdominoperineal excision (APE) was performed more often in underweight, obese class II and obese class III patients compared to normal weight, overweight, and obese class I patients, but did not reach significance (*p* = 0.074). Group 1 showed a lower rate of APR compared to Group 2 (15.3 vs. 31%, *p* = 0.025). A primary anastomosis was conducted in 82.5% with lower count in underweight, obese class II and obese class III individuals (*p* = 0.019) without reaching significant differences for anastomotic leakage (*p* = 0.466, 4.3% overall). General postoperative morbidity was increased in underweight, obese class II and obese class III patients compared to normal weight, overweight and obese class I individuals (*p* = 0.008). Surgical complications were significantly increased in the same patient collective (*p* = 0.006), whereas non-surgical complications showed no differences for BMI (*p* = 0.460). The postoperative 30-day mortality was 1.0% and did not depend on BMI (*p* = 0.116). Further data on quality indicators is shown in Table 3.

### 2.3. Local Recurrence

The 5-year rate for local recurrence in the entire study group was 5.9% (95% CI 3.9–7.9). Neither localisation of the tumour nor type of surgery or CEA-level had a significant influence on the 5-year local recurrence. Obesity class II and the presence of perioperative risk factors (ASA 3 and 4) showed a significant increment for local recurrence (Table 4).

### 2.4. Distant Metastasis

Distant metastases occurred in 16.7% (95% CI 13.6–19.8) of all patients in a 5-year observational period. Patients in Group 2 showed a rate of metastatic disease in 31.4% (95% CI 14.3–48.5) of cases while patients in Group 1 showed a significantly lower rate of 15.9% (95 CI 12.8–19.0; *p* = 0.034) (Figure 1). Significant differences between the BMI Groups as defined by the WHO did not occur.

Type of surgery, the preoperative ASA-Score and localisation of the tumour had no significant influence. Individuals with preoperative CEA levels over 5 ng/L had a significantly higher rate of distant metastases and an increased risk to develop distant metastases (Table 5).

### 2.5. Overall Survival

The 5-year overall survival was analyzed for 598 patients with an overall survival rate of 82.3% (CI 79.2–85.4). Patients in Group 1 (BMI 18.5–35 kg/m^2^) showed an overall survival of 82.9% (95% CI 79.8–86.0), which was significantly higher than in Group 2 (<18.5 or ≥35 kg/m^2^) with 70.0% (95% CI 53.5–86.5; *p* = 0.024) (Figure 2). No difference was observed between the BMI Groups as suggested by the WHO.

The localisation of the tumour had no significant impact, however APE was associated with a significantly lower survival rate. The presence of systemic disabilities, as classified by the ASA-Score, influenced the overall survival. Patients with an ASA-Score of 3 or 4 had a significantly lower 5-year-survial in this period of time. Elevated preoperative CEA-levels were also related to a lower 5-year survival and an increased mortality risk (Table 6).

## 3. Discussion

In the past years BMI has been proven to be a significant risk factor, or at least a surrogate parameter, for the development of colorectal cancer by different studies [8,10,11,12]. However, the evidence for BMI being a predictor of outcome in patients with rectal cancer undergoing surgical therapy is still discussed controversially.

Various trials have shown that underweight or obesity are risk factors for reduced survival in colorectal cancer diseases [13,14,15,16,17,18]. Therefor we decided to search for an indicator in these patients by dividing our patient cohort into two groups (Group 1 BMI 18.5–35 kg/m^2^; Group 2 BMI <18.5 and ≥35 kg/m^2^).

### 3.1. Quality Indicators

We detected differences in the rate of abdominoperineal excision (APE) between tumors in the low and middle third (45.6% vs. 3.9%, *p*-value < 0.001) of the rectum, but we were not able to confirm the disparate use of APE by sex, with a decreased rate among female patients with rectal cancer, as reported by a previous survey [17]. Patients included in our study, treated by APE, had a reduced survival compared to those treated by other surgical procedures (APE vs. other procedures: 73.0% vs. 83.9%; HR 2.0, *p*-value < 0.001), however, we did not see differences for distant metastases (18.0% vs. 16.4%, *p*-value 0.313) or local recurrence (8.9% vs. 5.4%, *p*-value 0.104). These findings are concordant to a study on 1598 patients with low and mid rectal cancer treated in 38 hospitals being part of the Spanish Rectal Cancer Project [18]. One possible hypothesis is that different patient characteristics, rather than the surgical procedure (APE) itself, cause a difference in oncological outcome in patients undergoing APE [19].

Postoperative complications were seen in 22.2% of all patients with a significantly higher percentage in underweight and obese class II and III patients. Previous studies have shown that morbidity influences the overall survival (OS) in patients with various tumor diseases [20,21,22]. We were able to confirm these results in our study (OS Morbidity vs. no Morbidity: 76.5% vs. 83.9%, *p*-value 0.004). No differences in non-surgical complications could be observed, indicating the differences in morbidity are caused by perioperative complications like wound dehiscence, anastomotic insufficiency or ileus, as reported in other studies [23,24,25]. One major factor leading to reduced overall survival in patients with postoperative complications is the reduced ability to receive adjuvant treatment [26,27,28]. Interestingly, we did not see any differences in adjuvant treatment rates between the BMI-groups, which leaves the mechanism behind the observed differences in the overall survival unanswered. 

### 3.2. Local Recurrence (LR)

Tumor localization in the rectum is known to affect the risk for the development of LR after primary curative surgical resection in rectal cancer patients [29]. Reduced surgical visibility in obese patients with lower rectal tumors compromising sufficient resection, are possible responsible factors. Despite these findings, patients included in our study did not show significantly higher rates of LR dependent on tumor localization, which might be correlated to the surgical procedure. APE has been proven to be a significant risk factor for LR in patients with rectal carcinoma [30]. Our patient cohort did not show the same dependency. Localization in the lower rectum or advanced tumor stage are common reasons to perform APE. While anatomy and tumor localization may be reasons for increased recurrences, tumor biology is another explanation worth considering. Obesity is associated with insulin resistance and consequently higher rates of circulating insulin [31] leading to a higher bioactivity of Insulin-like-growth factor 1 (IGF-1). IGF-1 promotes a series of intracellular signaling cascades leading to mitogenic and antiapoptotic events, a risk factor for recurrence and tumor growth [32,33]. No relation was seen between LR and tumor localization, whereas an advanced tumor stage increased LR risk.

### 3.3. Distant Metastasis (DM)

We were able to show a correlation of DM-rates by BMI and the UICC-stage. Our study population displayed increased rates of DM in patients with high or very low BMI (<18.5 or ≥35 kg/m^2^). The disparate appearance of DM might be due to molecular mechanisms as suggested by different studies pointing on inflammatory cytokines such as insulin-like growth factor-receptor (IGF-R) and leptin [34,35,36,37]. With obesity inducing a state of slight inflammation and changing the microenvironment for example in steatotic livers, this might be a reason for higher rates of DM in very obese patients [38]. Unfortunately, we have not been able to confirm this hypothesis due to missing blood samples for the patients included within this analysis. In underweight patients instead, the earlier infiltration of blood and lymph vessels can be seen as a factor for higher DM-rates. 

The possibility to provide adjuvant tumor therapy for patients with DM was diverging due to BMI (Group 1 vs. Group 2: 69.8% vs. 60.0%, *p*-value 0.036). This may be an effect of comorbidities in patients with high and very low BMI.

### 3.4. Overall Survival

Patients with advanced tumors are known to have a decreased time of overall survival [39]. This circumstance is reflective of the advanced stage of the tumor-disease.

The results of our study suggest an association between BMI and overall survival and are consistent with a study performed on 526 patients, determining a link between BMI and patients outcome diagnosed with left-sided cancer and rectal cancer [40]. Reduced survival in patients with obesity class II or higher as well as underweight (Group 2) compared to normal weight, overweight and obesity class I (Group 1) individuals, may be related to higher rates of distant metastases (DM) found in this group. At least the lower survival rate in patients with underweight might be derived from an advanced tumor stage causing a catabolic metabolism. Weight-loss promoted by this catabolic situation could be a factor shifting patients with normal weight to underweight at the time of hospital admission [9]. Chemotherapy-related toxicities and other complications had been observed in slightly lower numbers in obese patients compared to non-obese patients in some adjuvant therapy trials [14,41]. The possibility to provide appropriate postoperative tumor therapy can be suggested as a factor for better survival in obese patients.

Tumor localization in contrast to other trials, had no significant impact on the overall survival of patients included in this study [42]. This may be reflective of a good surgical quality independent of difficult conditions.

The strength of our study is the large sample size, stage specific data and the long-term follow-up data with at least 5-years follow up-time for each patient, as well as completeness of the data. On the other hand, this study has certain limitations: we performed a retrospective analysis of prospectively collected data. The small number of patients in the subgroups as well as the design as a single center study reduced the validity of our conclusions. 

## 4. Materials and Methods

The initially screened study cohort consisted of patients diagnosed with rectal cancer at the University Hospital Erlangen in the time of January 2003 until December 2010. All of these patients underwent surgery for the treatment of rectal carcinoma in the department of surgery at the University Hospital Erlangen. Patient data were selected on the following inclusion criteria: Solitary invasive rectal carcinoma (at least infiltration of the submucosa) ≤16 cm from anal verge when measured with a rigid sigmoidorectoscope; treated by (low) anterior resection, Hartmann procedure, intersphincteric resection or abdominoperineal excision; no other malignant tumors either synchronous with or prior to diagnosis; no history of familial adenomatous polyposis, ulcerative colitis or Crohn’s disease; no distant metastasis (M0) at the time of diagnosis. 6 patients without valid BMI, 6 patients who died postoperatively and three patients with unknown tumor-status were excluded from the analysis. 

Only patients with microscopic curative resection (R0) were used for outcome analyses. Consequently, a total of 598 patients were included for these analyses.

To calculate surgical quality indicators like R0-resection rate in Table 3 we included all patients independent of their resection status (R0 (*n* = 598) + R1 (*n* = 9) + R2 (*n* = 5) = 612).

Long-term chemoradiation was administered in the majority of patients after preoperative staging (cT3, 4 or cN+ [43]), in concordance to the protocol of the German CAO/ARO/AIO-94 study [44]. The neoadjuvant treatment was finished 6–8 weeks prior to surgery. 

All samples were categorized according to the seventh edition of the UICC TNM classification [45].

According to the definition of the World Health Organization (WHO) the patient cohort was divided in six groups, i.e., underweight (BMI < 18.5 kg/m^2^; *n* = 5); normal weight (BMI 18.5–24,9 kg/m^2^; *n* = 209); overweight (BMI 25.0–29.9 kg/m^2^; *n* = 257); obesity class I (BMI 30.0–34.9 kg/m^2^; *n* = 102); obesity class II (BMI 35.0–39.9 kg/m^2^; *n* = 16); and obesity class III (BMI ≥ 40.0 kg/m^2^; *n* = 9) [46]. BMI was measured in the anaesthesia outpatient clinic during preparation for surgery at the time of first hospital admission at the University Hospital Erlangen.

To meet the challenge of a society with an increasing prevalence of obesity in the last 30 years [47], we divided the study population in two groups (Group 1 BMI 18.5– 35 kg/m^2^; *n* = 568; Group 2 BMI <18.5 and ≥35 kg/m^2^; *n* = 30) with Group 1 representing the dominant phenotype in future communities.

Patients with tumors in the lower and middle rectum received a total mesorectal excision (TME). Partial mesorectal excision (PME) or TME was performed if suitable in patients with tumour in the upper part of the rectum. The quality of TME or PME was examined in accordance to the protocol of Quirke et al. by a pathologist [48]. 

All patients were followed up with for at least 5 years: twice a year, for the first 2 years and then yearly for the remaining period as suggested by the first edition of the German S3-Guidelines for Colorectal Carcinoma [43]. Follow-ups included physical examination, analysis of carcinoembryonic antigen (CEA) levels, chest X-ray, abdominal ultrasonography, computed tomography of the pelvis and rectoscopy. 

Follow-up data were either collected by standard follow-up examinations at the University hospital Erlangen or by written correspondence with the patient’s general practitioner. Subsequently the vital status of the patient was validated through inquiries at the patient’s local registration office and all data have been collected and categorized by our local cancer registry.

All subjects gave their informed consent for inclusion before they participated in this study. The study was conducted in accordance with the Declaration of Helsinki, and the protocol was approved by the Ethics Committee of the Friedrich-Alexander-Universität Erlangen-Nürnberg, Germany (172_19 Bc).

### Statistical Analysis

To compare categorical data χ^2^ and Fisher’s exact test were used. For quantitative data the Mann-Whitney *U* test was applied. The Kaplan-Meier method was utilized to analyze the five-year rates of local recurrence, distant metastasis and overall survival; the log-rank test was applied for comparison of the survival curves. For overall survival death from any cause was defined as an event. To analyze univariable and multivariable differences in local recurrence, distant metastasis or overall survival, a Cox-regression model was used. A *p*-value < 0,050 was considered to be significant. 

The statistical software package SPSS^®^ version 24 (IBM, Armonk, NY, USA) was used for all analyses.

## 5. Conclusions

We were able to show differences in survival rates depending on BMI values. The increased rate of distant metastases as well as the higher rates of APE in Group 2 might be due to patient selection in our cohort. Patients treated by APE had a 2-fold risk of death, but no elevated rates of DM. LR increased the rate of DM by a factor of 3,6, but showed no impact on overall survival. 

## Figures and Tables

**Figure 1 cancers-11-00609-f001:**
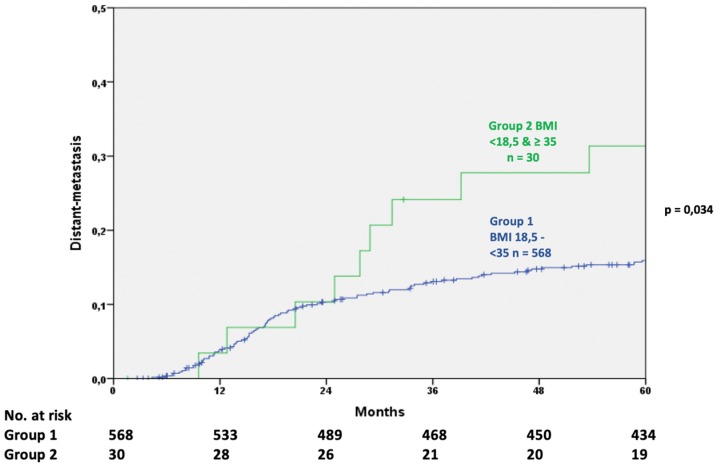
Comparison of distant metastasis after oncological treatment (log rank test).

**Figure 2 cancers-11-00609-f002:**
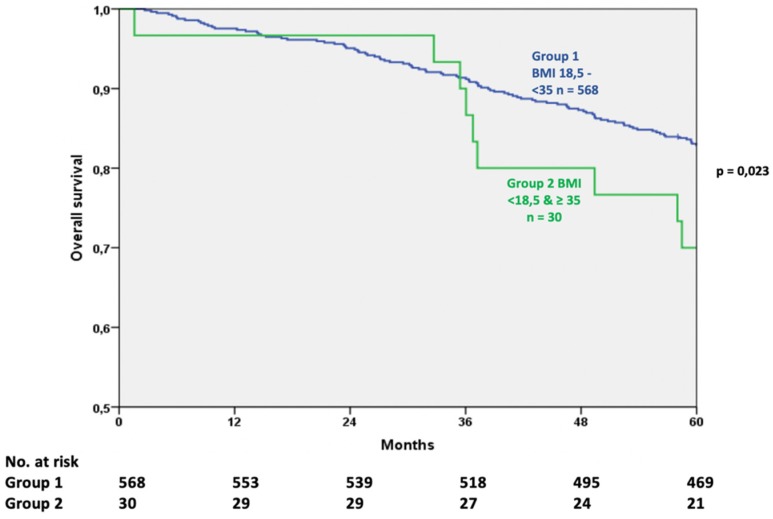
Comparison of overall survival after oncological treatment (log rank test).

**Table 1 cancers-11-00609-t001:** Demographics, tumour and treatment characteristics of 612 patients.

Patient Characteristics	Number	Percent
Age (years), median (range)	65 (18–91)	
Sex		
Male	406	66.3
Female	206	33.7
BMI Groups		
Group 1 BMI 18.5–35	580	94.8
Group 2: BMI <18.5 & ≥35	32	5.2
BMI WHO		
<18.5 underweight	7	1.1
18.5–24.9 normal weight	212	34.6
25.0–29.9 overweight	266	43.5
30.0–34.9 class I obesity	102	16.7
35.0–39.9 class II obesity	16	2.6
≥40.0 class III obesity	9	1.5
Tumour site		
<6 cm lower third	211	34.5
6–12 cm middle third	276	45.1
12–16 cm upper third	125	20.4
ASA-Score		
ASA 1 & 2	490	80.1
ASA 3 & 4	118	19.3
Missing	4	0.7
Pretherapy CEA level		
normal(<5 ng/L)	438	71.6
raised(>10 ng/L)	94	15.4
Missing	80	13.1
Neoadjuvant therapy		
neoadjuvant RCT	320	52.3
neoadjuvant RT	4	0.7
neoadjuvant CT	1	0.2
None	287	46.9
Elective surgery		
Yes	606	99.0
No, Emergency	6	1.0
Type of surgery		
Anterior resection	77	12.6
Low anterior resection	378	61.8
Hartmann procedure	8	1.3
Intersphincteric resection	50	8.2
Abdominoperineal excision	99	16.2
Multivisceral resection		
No	520	85.0
Yes	92	15.0
Postoperative therapy		
None	70	11.4
CT	171	27.9
RT	4	0.7
RCT	56	9.2
unknown	311	50.8
pT-category		
pT1	66	10.8
pT2	103	16.8
pT3	100	16.3
pT4	18	2.9
ypT0	60	9.8
ypT1	13	2.1
ypT2	113	18.5
ypT3	120	19.6
ypT4	19	3.1
pN-category		
pN0	217	35.5
pN1	46	7.5
pN2	24	3.9
ypN0	235	38.4
ypN1	69	11.3
ypN2	21	3.4
UICC-stage		
I	150	24.5
II	67	10.9
III	70	11.4
y0	55	9.0
yI	101	16.5
yII	79	12.9
yIII	90	14.7

ypT: pathological tumour category after chemoradiation, ypN: pathological node category after chemoradiation, BMI: Body Mass Index, WHO: World Health Orga, ASA: American Society of Anaesthesiologists, CEA: Carcinoembryonic antigen, RCT: Radiochemotherapy, CT: Chemotherapy, UICC: Union internatonale contre le cancer.

**Table 2 cancers-11-00609-t002:** Demographics, tumour and treatment characteristics of 612 patients for BMI-groups.

Patient Characteristics	<18.5 kg/m^2^ Underweight (*n* = 7)	18.5–25 kg/m^2^ Normal Range (*n* = 212)	25–30 kg/m^2^ Overweight (*n* = 266)	30–35 kg/m^2^ Obese Class I (*n* = 102)	35–40 kg/m^2^ Obese Class II (*n* = 16)	≥40 kg/m2 Obese Class III (*n* = 9)	*p*	Group 1: BMI 18.5–35 (*n* = 580)	Group 2: BMI <18.5 & ≥35 (*n* = 32)	*p*
Age										
< 65	6 (86)	118 (55.7)	137 (51.5)	45 (44.1)	8 (50)	6 (67)		300 (51.7)	20 (63)	
≥ 65	1 (14)	94 (44.3)	129 (48.5)	57 (55.9)	8 (50)	3 (33)	0.180	280 (48.3)	12 (37)	0.277
Sex										
Male	3 (43)	122 (57.5)	196 (73.7)	69 (67.6)	11 (69)	5 (56)		387 (66.7)	19 (59)	
Female	4 (57)	90 (42.5)	70 (26.3)	33 (32.4)	5 (31)	4 (44)	0.005	193 (33.3)	13 (41)	0.443
Tumour site										
<6cm lower third	4 (57)	83 (39.2)	82 (30.8)	34 (33.3)	5 (31)	3 (33)		199 (34.3)	12 (38)	
<12 cm middle third	3 (43)	84 (39.6)	132 (49.6)	43 (42.2)	9 (56)	5 (56)		259 (44.7)	17 (53)	
≤16cm upper third	0 (0)	45 (21.2)	52 (19.6)	25 (24.5)	2 (13)	1 (11)	0.416	122 (21.0)	3 (9)	0.269
ASA-Score ^a^										
ASA 1&2	4 (57)	181/211 (85.8)	215/264 (81.4)	75/101 (74.3)	9 (56)	6 (67)		471/576 (81.8)	19 (59)	
ASA 3&4	3 (43)	30/211 (14.2)	49/264 (18.6)	26/101 (25.7)	7 (44)	3 (33)	0.005	105/576 (18.2)	13 (41)	0.004
Pretherapy CEA level ^b^										
Normal (<5 ng/L)	3/6 (50)	143/176 (81.3)	196/231 (84.8)	78/94 (83.0)	9 (56)	9 (100)		417/501 (83.2)	21/31 (68)	
Elevated (≥5 ng/L)	3/6 (50)	33/176 (18.7)	35/231 (14.2)	16/94 (17.0)	7 (44)	0 (0)	0.018	84/501 (16.8)	10/31 (32)	0.048
Neoadjuvant therapy										
Neoadjuvant RCT	6 (86)	117 (55.2)	133 (50.0)	52 (51.0)	9 (56)	9 (47)		302 (52.1)	18 (56)	
Neoadjuvant RT	0 (0)	1 (0.5)	3 (1.1)	0 (0.0)	0 (0)	0 (0)		4 (0.7)	0 (0)	
Neoadjuvant CT	0 (0)	0 (0.0)	1 (0.4)	0 (0.0)	0 (0)	0 (0)		1 (0.1)	0 (0)	
None	1 (14)	94 (44.3)	129 (48.5)	50 (49.0)	7 (44)	10 (53)	0.635	273 (47.1)	14 (44)	0.786
Elective surgery										
Yes	7 (100)	208 (98.1)	264 (99.2)	102 (100)	16 (100)	9 (100)		574 (99.0)	32 (100)	
No, Emergency	0 (0)	4 (1.9)	2 (0.8)	0 (0.0)	0 (0)	0 (0)	0.523	6 (1.0)	0 (0)	1.0
Type of surgery										
Anterior resection	0 (0)	32 (15.1)	28 (10.5)	15 (14.7)	1 (6)	1 (11)		75 (12.9)	2 (6)	
Low anterior resection	2 (29)	126 (59.4)	177 (66.5)	59 (57.8)	9 (56)	5 (56)		362 (62.4)	16 (50)	
Hartmann procedure	1 (14)	3 (1.4)	3 (1.1)	1 (1.1)	0 (0)	0 (0)		7 (1.3)	1 (3)	
Intersphincteric resection	0 (0)	15 (7.1)	19 (7.1)	13 (12.7)	2 (13)	1 (11)		47 (8.1)	3 (9)	
Abdominoperineal excision	4 (57)	36 (17.0)	39 (14.8)	14 (13.7)	4 (25)	2 (22)	0.863	89 (15.3)	10 (32)	0.086
Multivisceral resection										
No	3 (43)	181 (85.4)	228 (85.7)	88 (86.3)	13 (81)	7 (78)		497 (85.7)	23 (72)	
Yes	4 (57)	31 (14.6)	38 (14.3)	14 (13.7)	3 (19)	2 (22)	0.098	83 (14.3)	9 (28)	0.042
Postoperative therapy ^c^										
RCT	0/6 (0)	19/105 (18.1)	26/127 (20.5)	8/49 (16.3)	1/9 (11)	2/5 (40)		53/281 (18.9)	3/20 (15)	
Radiotherapy	0/6 (0)	0/ 105 (0.0)	2/127 (1.6)	2/49 (4.2)	0/9 (0)	0/5 (0)		4/281 (1.4)	0/20 (0)	
Chemotherapy	5/6 (83.3)	55/105 (52.4)	77/127 (60.6)	28/49 (57.1)	5/9 (56)	1/5 (20)		160/281 (56.9)	11/20 (55)	
None	1/6 (16.7)	31/105 (29.5)	22/127 (17.3)	11/49. (22.4)	3/9 (33)	2/5 (40)	0.320	64/281 (22.8)	6/20 (30)	0.856
pT-category										
pT1	0 (0)	22 (10.4)	25 (9.4)	14 (13.7)	2 (13)	3 (34)		61 (10.5)	5 (16)	
pT2	0 (0)	31 (14.6)	49 (18.4)	20 (19.6)	2 (13)	1 (11)		100 (17.2)	3 (9)	
pT3	1 (14)	34 (16.0)	45 (16.9)	15 (14.7)	3 (19)	2 (22)		94 (16.2)	6 (19)	
pT4	0 (0)	7 (3.3)	10 (3.8)	1 (1.1)	0 (0)	0 (0)		18 (3.2)	0 (0)	
ypT0	0 (0)	22 (10.4)	29 (10.9)	7 (6.9)	1 (6)	1 (11)		58 (10.0)	2 (6)	
ypT1	0 (0)	5 (2.4)	7 (2.6)	1 (1.1)	0 (0)	0 (0)		13 (2.2)	0 (0)	
ypT2	1 (14)	46 (21.7)	45 (16.9)	18 (17.5)	3 (19)	0 (0)		109 (18.8)	4 (13)	
ypT3	3 (43)	39 (18.4)	48 (18.0)	24 (23.5)	4 (24)	2 (22)		111 (19.1)	9 (28)	
ypT4	2 (29)	6 (2.8)	8 (3.1)	2 (1.9)	1 (6)	0 (0)	0.110	16 (2.8)	3 (9)	0.333
pN-category										
pN0	1 (14)	67 (31.6)	100 (37.6)	42 (41.2)	4 (25)	3 (33)		209 (36.0)	8 (25)	
pN1	0 (0)	16 (7.5)	20 (7.5)	6 (5.9)	2 (13)	2 (23)		42 (7.2)	4 (13)	
pN2	0 (0)	11 (5.2)	9 (3.4)	2 (2.0)	1 (6)	1 (11)		22 (3.8)	2 (6)	
ypN0	3 (43)	90 (42.5)	102 (38.3)	30 (29.4)	7 (44)	3 (33)		222 (38.3)	13 (41)	
ypN1	2 (29)	21 (9.9)	26 (9.8)	19 (18.6)	1 (6)	0 (0)		66 (11.4)	3 (9)	
ypN2	1 (14)	7 (3.3)	9 (3.4)	3 (2.9)	1 (6)	0 (0)	0.140	19 (3.3)	2 (6)	0.408

^a^ 4 Patients excluded due to unknown ASA-Score, ^b^ 80 Patients excluded due to unknown CEA-level, ^c^ 311 patients excluded due to missing data on postoperative therapy, Values in parentheses are percentages.

**Table 3 cancers-11-00609-t003:** Surgical quality indicators for 612 patients.

Quality Indicators	<18.5 Underweight *n* = 7)	18.5–25 Normal Weight (*n* = 212)	25–30 Overweight (*n* = 266)	30–35 Obese Class I (*n* = 102)	35–40 Obese Class II (*n* = 16)	≥40 Obese Class III (*n* = 9)	*p*	Group 1: 18.5–35 (*n* = 580)	Group 2: <18.5 & ≥ 35 (*n* = 32)	*p*
Number of examined LN; R0, R1, R2 *n* = 612 ^a^, median (range)	13 (11–33)	20 (2–57)	21.5 (3–62)	21 (5–76)	23.5 (8–57)	30 (15–43)		21 (2–76)	25.5 (8 – 57)	
≥ 12 LN examined	6/7 (86)	188/212 (88.7)	239/266 (89.8)	93/102 (91.2)	15/16 (94)	9/9 (100)	0.828	520/580 (89.7)	30/32 (94)	0.742
pN0 *n* = 217, median (range)	33 (33–33)	24 (12–51)	27 (7–57)	31 (10–76)	30 (22–36)	26 (15–41)		26 (7–76)	30 (15–41)	
≥ 12 LN examined	1/1 (100)	67/67 (100)	98/100 (98.0)	41/42 (97.6)	4/4 (100)	3/3 (100)	0.664	206/209 (98.6)	8/8 (100)	0.737
ypN0 *n* = 235, median (range)	13 (12–13)	17 (2–36)	18 (3–62)	17 (7–37)	21 (8–29)	41 (16–43)		17.5 (2–62)	16 (8–43)	
≥ 12 LN examined	3/3 (100)	70/90 (77.8)	79/102 (77.5)	26/30 (86.7)	6/7 (86)	3/3 (100)	0.004	175/222 (78.8)	12/13 (92)	0.238
R0—resection-rate	5/7 (71)	209/212 (98.6)	257/266 (96.6)	102/102 (100)	16/16 (100)	9/9 (100)	0.031	568/580 (97.9)	30/32 (94)	0.163
Aboral resection margin > 1 mm	7/7 (100)	210/212 (99.1)	263/266 (98.9)	100/102 (98.0)	15/16 (94)	9/9 (100)	0.474	573/580 (98.8)	31/32 (97)	0.351
Circumferential resection margin > 1 mm	5/7 (71)	197/212 (92.9)	244/266 (91.7)	98/102 (96.1)	15/16 (94)	9/9 (100)	0.186	539/580 (92.9)	29/32 (91)	0.096
Intraoperative local tumour cell dissemination	2/7 (29)	10/212 (4.7)	14/266 (5.3)	2/102 (2.0)	1/ 16 (6)	0/9 (0)	0.112	26/580 (4.5)	3/32 (9)	0.279
Quality of TME/PME ^b^										
Mesorectal/intramesorectal plane	6/6 (100)	190/196 (96.9)	246/253 (97.2)	91/92 (98.9)	14/15 (93,3)	8/9 (88,9)		527/541 (97.4)	28/30 (93)	
Muscularis propria plane	0/6 (0)	6/196 (3.1)	7/253 (2.8)	1/92 (1.1)	1/15 (6,7))	1/9 (11,1)	0.309	14/541 (2.6)	2/30 (7)	0.203
Abdominoperineal excision										
All locations	4/7 (57)	36/212 (17.0)	39/266 (14.7)	14/102 (13.7)	4/16 (25)	2/9 (23)	0.074	89/580 (15.3)	10/32 (31)	0.025
<6 cm lower third	4/4 (100)	30/83 (36.1)	30/82 (36.6)	14/34 (41.2)	4/5 (80)	2/3 (67)	<0.001	74/199 (37.2)	10/12 (83)	<0.001
6–12 cm middle third	0/3 (0)	6/84 (7.1)	9/132 (6.8)	0/43 (0)	0/9 (0)	0/5 (0)	<0.001	15/259 (5.8)	0/17 (0)	<0.001
Anastomosis	2/7 (29)	173/212 (81.6)	224/266 (84.2)	87/102 (85.3)	12 /16 (75)	7/9 (78)	0.019	484/580 (83.4)	21/32 (66)	0.016
Anastomotic leak	0/2 (0)	5/173 (2.9)	10/224 (4.5)	7/87 (8)	0/12 (0)	0/7 (0)	0.466	22/484 (4.5)	0/21 (0)	1.0
Morbidity										
Total	2/7 (29)	39/212 (18.4)	55/266 (20.7)	29/102 (28.4)	9/16 (56)	3/9 (33)	0.008	123/580 (21.2)	14/32 (44)	0.005
Non-surgical	0/7 (0,)	9/212 (4.2)	12/266 (4.5)	7/102 (6.8)	1/16 (6)	0/9 (0)	0.460	26/580 (4.5)	1/32 (3)	1.0
Surgical	2/7 (29)	30/212 (14.1)	43/266 (16.2)	22/102 (21.6)	8/16 (50)	3/9 (33)	0.006	95/580 (16.4)	13/32 (41)	0.001
Postoperative 30-day mortality	1/7 (14)	3/212 (1.4)	1/266 (0.4)	1/102 (1.0)	0/16 (0)	0/9 (0)	0.116	5/580 (0.9)	1/32 (3)	0.282

^a^ R0 = 598 patients + R1 = 9 patients + R2 = 5 patients = 612 patients; ^b^ 41 patients missing due to unknown data.

**Table 4 cancers-11-00609-t004:** 5-year rate of locoregional recurrence for 598 patients.

Univariate Analysis					Multivariate Analysis		
Patient and Tumour Characteristics	Number	5Y-locoregional Recurrence	95% CI	*p*	Hazard Ratio	95% CI	*p*
Total	598	5.9%	3.9–7.9				
BMI WHO							
BMI < 18.5 kg/m^2^ underweight	5	0%			0.0		
BMI 18.5–25 kg/m^2^ normal range	209	3.7%	1.0–6.4		1.0		
BMI 25–30 kg/m^2^ overweight	257	8.7%	5.2–12.2		1.7	0.8–3.7	0.150
BMI 30–35 kg/m^2^ obese class I	102	1.1%	0.0–3.3		0.6	0.2–1.9	0.357
BMI 35–40 kg/m^2^ obese class II	16	23.6%	0.0–47.3		4.1	1.1–15.4	0.036
BMI ≥ 40 kg/m^2^ obese class III	9	11.1%	0.0–31.7	0.112	2.0	0.2–16.3	0.533
BMI Groups							
Group 1: BMI 18.5–35	568	5.5%	3.5–7.5		1.0		
Group 2: BMI <18.5 & ≥35	30	15.4%	1.5–29.3	0.106	2.0	0.7–5.9	0.187
Age							
≤65	315	5.6%	3.1–8.1				
>65	283	6.4%	3.5–9.3	0.884			
Sex							
Male	394	6.7%	4.2–9.2				
Female	204	4.3%	1.4–7.2	0.111			
Localisation							
<6 cm	202	5.3%	2.2–8.4				
6–12 cm	271	7.1%	4.0–10.2				
12–16 cm	125	4.3%	0.6–8.0	0.641			
UICC-stage							
I	150	4.3%	1.0–7.6		1.0		
II	65	11.4%	3.4–19.4		3.8	1.4–10.0	0.007
III	65	9.2%	1.4–17.0		1.9	0.6–6.1	0.258
y0	55	0%			0		0.973
yI	101	3.0%	0–6.3		0.6	0.2–2.4	0.487
yII	74	7.5%	1.0–14.0		1.7	0.6–5.2	0.321
yIII	88	8.9%	2.6–15.2	0.003	2.5	0.9–6.8	0.067
Abdominoperineal excision							
No	509	5.4%	3.4–7.4				
Yes	89	8.9%	2.6–15.2	0.104			
ASA-Score ^a^							
ASA 1&2	481	5.0%	3.0–7.0		1.0		
ASA 3&4	113	10.6%	4.3–16.9	0.040	2.1	1.0–4.5	0.042
CEA-level ^b^							
Normal (<5 ng/L)	432	5.9%	3.5–8.3		1.0		
Elevated (≥5 ng/L)	88	4.9%	0.2–9.6	0.389	1.9	0.9–3.9	0.075

^a^ 4 Patients excluded due to unknown ASA-Score, ^b^ 78 Patients excluded due to unknown CEA-level.

**Table 5 cancers-11-00609-t005:** 5-year rate of distant metastases of 598 patients.

Univariate Analysis					Multivariate Analysis		
Patient and Tumour Characteristics	Number	5Y distant Metastases	95% CI	*p*	Hazard Ratio	95% CI	*p*
Total	598	16.7%	13.6–19.8				
BMI WHO							
BMI < 18.5 kg/m^2^ underweight	5	40.0%	0–82.9		1.7	0.4–7.2	0.493
BMI 18.5–25 kg/m^2^ normal range	209	16.2%	11.1–21.3		1.0		
BMI 25–30 kg/m^2^ overweight	257	17.3%	12.6–22.0		1.2	0.8–2.0	0.340
BMI 30–35 kg/m^2^ obese class I	102	11.9%	5.6–18.2		0.9	0.5–1.7	0.813
BMI 35–40 kg/m^2^ obese class II	16	34.0%	9.7–58.3		1.9	0.8–4.8	0.153
BMI ≥ 40 kg/m^2^ obese class III	9	22.2%	0–49.4	0.225	1.4	0.3–5.8	0.678
BMI Groups							
Group 1: BMI 18.5–35	568	15.9%	12.8–19.0		1.0		
Group 2: BMI <18.5 or ≥35	30	31.4%	14.3–48.5	0.034	1.6	0.8–3.1	0.177
Age							
≤65	315	14.1%	10.2–18.0				
>65	283	19.7%	14.8–24.6	0.085			
Sex							
Male	394	16.3%	12.6–20.0				
Female	204	17.4%	12.1–22.7	0.861			
Localisation							
<6 cm	202	13.8%	9.0–18.8				
6–12 cm	271	17.9%	13.2–22.6				
12–16 cm	125	18.8%	11.7–25.9	0.541			
UICC-stage							
I	150	9.8%	4.9–14.7		1.0		
II	65	23.1%	12.5–33.7		2.1	1.0–4.4	0.041
III	65	30.7%	18.4–43.0		2.7	1.4–5.5	0.003
y0	55	1.9%	0–5.4		0.2	0–1.3	0.086
yI	101	6.9%	2.0–11.8		0.6	0.2–1.4	0.250
yII	74	23.3%	13.7–32.9		2.3	1.1–4.6	0.019
yIII	88	30.3%	20.5–40.1	<0.001	3.1	1.7–5.8	<0.001
Abdominoperineal excision							
No	509	16.4%	13.1–19.7				
Yes	89	18.0%	9.8–26.2	0.313			
ASA-Score ^a^							
ASA 1&2	481	15.2%	11.9–18.5				
ASA 3&4	113	24.2%	15.8–32.6	0.052			
CEA-level ^b^							
Normal (<5 ng/L)	432	14.1%	10.8–17.4		1.0		
Elevated(≥5 ng/L)	88	31.6%	21.6–41.6	<0.001	1.9	1.2–3.0	0.005

^a^ 4 Patients excluded due to unknown ASA-Score, ^b^ 78 Patients excluded due to unknown CEA-level.

**Table 6 cancers-11-00609-t006:** 5-year overall survival for 598 patients.

Univariate Analysis					Multivariate Analysis		
Patient and Tumour Characteristics	Number	5-OSR	95%CI	*p*	Hazard Ratio	95%CI	*p*
Total	598	82.3%	79.2–85.4				
BMI WHO							
BMI < 18.5 kg/m^2^ underweight	5	60.0%	17.1–100		1.7	0.4–7.2	0.458
BMI 18.5–25 kg/m^2^ normal range	209	82.3%	77.2–87.4		1.0		
BMI 25–30 kg/m^2^ overweight	257	81.3%	76.6–86.0		1.2	0.8–1.7	0.369
BMI 30–35 kg/m^2^ obese class I	102	88.2%	81.9–94.5		0.7	0.5–1.2	0.219
BMI 35–40 kg/m^2^ obese class II	16	68.8%	46.1–91.5		1.4	0.7–2.9	0.377
BMI ≥ 40 kg/m^2^ obese class III	9	77.8%	50.6–100	0.211	0.7	0.2–2.0	0.524
BMI Groups							
Group 1: BMI 18.5–35	568	82.9%	79.8–86.0		1.0		
Group 2: BMI <18.5 or ≥35	30	70.0%	53.5–86.5	0.023	1.1	0.7–1.9	0.652
Age							
≤65	315	90.8%	87.7–93.9				
>65	283	72.8%	67.7–77.9	<0.001			<0.001
Sex							
Male	394	83.5%	79.8–87.2				
Female	204	79.9%	74.4–85.4	0.471			
Morbidity							
Yes	132	76.5%	69.2–83.8				
No	466	83.9%	80.6–87.2	0.004			
Localisation							
<6 cm	202	85.1%	80.2–90.0				
6–<12 cm	271	80.4%	75.7–85.1				
12–16 cm	125	81.6%	74.7–88.5	0.176			
UICC-stage							
I	150	84.7%	79.0– 90.4		1.0		
II	65	73.8%	63.0–84.6		1.2	0.7–2.0	0.476
III	65	63.1%	51.3–74.9		2.3	1.5–3.6	<0.001
y0	55	98.2%	94.7–100		0.4	0.2–0.9	0.024
yI	101	97.0%	93.7–100		0.5	0.3–0.8	0.010
yII	74	74.3%	64.3–84.3		1.0	0.6–1.7	0.973
yIII	88	78.4%	69.8–87.0	<0.001	1.1	0.7– 1.8	0.658
Abdominoperineal excision							
No	509	83.9%	80.8–87.0		1.0		
Yes	89	73.0%	63.8–82.2	0.001	1.9	1.3–2.9	0.001
ASA-Score ^a^							
ASA 1&2	481	86.3%	83.2–89.4		1.0		
ASA 3&4	113	64.6%	55.8–73.4	<0.001	3.2	2.3–4.4	<0.001
Pretherapy CEA level ^b^							
Normal (<5 ng/mL)	432	84.3%	80.8–87.8		1.0		
Elevated (≥5 ng/mL)	88	68.2%	58.4–78.0	<0.001	1.9	1.3–2.6	<0.001

^a^ 4 Patients excluded due to unknown ASA-Score; ^b^ 78 Patients excluded due to unknown CEA-level.

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
