# Peer review of "Influence of Body Mass Index on Long-Term Outcome in Patients with Rectal Cancer—A Single Centre Experience"

_cancers, 2019, doi:10.3390/cancers11050609_

Round 1

Reviewer 1 Report

In the present study, Kalb and co-workers evaluated the influence of BMI on long-term outcomes in patients with rectal cancer. Authors demonstrated that BMI significantly influences the evaluated outcomes, including overall survival and rates of distant metastases and concluded that underweight and severe obesity are strongly associated with them, representing a further risk factor for this class of patients. The issue of this study is interesting and in line with the aims of the Journal. The study has been well conducted and correctly presented.

However, as minor concerns:

·         Line #21: is it correct to define cachexia only on the basis of BMI evaluation? Body composition assessment should be considered in order to establish the presence of cachexia. “Cachexia” should be replaced with “underweight” overall in the text

·         Line #40: this statement should be supported by proper references

·         According to the Journal guidelines for authors: “When reporting on research that involves human subjects, human material, human tissues, or human data, authors must declare that the investigations were carried out following the rules of the Declaration of Helsinki of 1975 (https://www.wma.net/what-we-do/medical-ethics/declaration-of-helsinki/), revised in 2013. According to point 23 of this declaration, an approval from an ethics committee should have been obtained before undertaking the research. At a minimum, a statement including the project identification code, date of approval and name of the ethics committee or institutional review board should be cited in the Methods Section of the article. Data relating to individual participants must be described in detail, but private information identifying participants need not be included unless the identifiable materials are of relevance to the research (for example, photographs of participants’ faces that show a particular symptom). Editors reserve the right to reject any submission that does not meet these requirements.

Example of an ethical statement: "All subjects gave their informed consent for inclusion before they participated in the study. The study was conducted in accordance with the Declaration of Helsinki, and the protocol was approved by the Ethics Committee of XXX (Project identification code)." “

·         As this topic is very current, it is advisable to support this manuscript with more recent references

In this referee’s opinion, with suggested revisions, the manuscript is suitable for publication on this Journal.

Author Response

R1: In the present study, Kalb and co-workers evaluated the influence of BMI on long-term outcomes in patients with rectal cancer. Authors demonstrated that BMI significantly influences the evaluated outcomes, including overall survival and rates of distant metastases and concluded that underweight and severe obesity are strongly associated with them, representing a further risk factor for this class of patients. The issue of this study is interesting and in line with the aims of the Journal. The study has been well conducted and correctly presented.

A: We thank this reviewer for the positive comments on our study.

R1: However, as minor concerns:

Line #21: is it correct to define cachexia only on the basis of BMI evaluation? Body composition assessment should be considered in order to establish the presence of cachexia. “Cachexia” should be replaced with “underweight” overall in the text:

A: We agree with this reviewer that cachexia cannot only be defined on the basis of BMI evaluation and therefore we have replaced the word "cachexia" with "underweight" overall in the text.

R1: Line #40: this statement should be supported by proper references:

A: We have included now two additional references supporting our statement. These are:

1. Deng T, Lyon CJ, Bergin S, et al (2016) Obesity, Inflammation, and Cancer. Annu Rev Pathol 11:421–449.

2. Ma Y, Yang Y, Wang F, et al (2013) Obesity and risk of colorectal cancer: a systematic review of prospective studies. PloS One 8:e53916.

R1: According to the Journal guidelines for authors: “When reporting on research that involves human subjects, human material, human tissues, or human data, authors must declare that the investigations were carried out following the rules of the Declaration of Helsinki of 1975 (https://www.wma.net/what-we-do/medical-ethics/declaration-of-helsinki/), revised in 2013. According to point 23 of this declaration, an approval from an ethics committee should have been obtained before undertaking the research. At a minimum, a statement including the project identification code, date of approval and name of the ethics committee or institutional review board should be cited in the Methods Section of the article. Data relating to individual participants must be described in detail, but private information identifying participants need not be included unless the identifiable materials are of relevance to the research (for example, photographs of participants’ faces that show a particular symptom). Editors reserve the right to reject any submission that does not meet these requirements.

Example of an ethical statement: "All subjects gave their informed consent for inclusion before they participated in the study. The study was conducted in accordance with the Declaration of Helsinki, and the protocol was approved by the Ethics Committee of XXX (Project identification code)." “

A: We thank this reviewer for this important comment. We have included now the name of the ethics committee and the project identification code. The ethics committee approved this particular retrospective analysis. The new paragraph in the manuscript reads as follows:

"All subjects gave their informed consent for inclusion before they participated in this study. The study was conducted in accordance with the Declaration of Helsinki, and the protocol was approved by the Ethics Committee of the Friedrich-Alexander-Universität Erlangen-Nürnberg, Germany (172_19 Bc)."

R1: As this topic is very current, it is advisable to support this manuscript with more recent references:

A: We have included now more recent references addressing the importance of this very current topic.

1.Lee J, Meyerhardt JA, Giovannucci E, Jeon JY (2015) Association between body mass index and prognosis of colorectal cancer: a meta-analysis of prospective cohort studies. PloS One 10:e0120706.

2.Yu H, Rohan T (2000) Role of the insulin-like growth factor family in cancer development and progression. J Natl Cancer Inst 92:1472–1489

3.Wang Z, Aguilar EG, Luna JI, et al (2019) Paradoxical effects of obesity on T cell function during tumor progression and PD-1 checkpoint blockade. Nat Med 25:141–151.

4.Clements VK, Long T, Long R, et al (2018) Frontline Science: High fat diet and leptin promote tumor progression by inducing myeloid-derived suppressor cells. J Leukoc Biol 103:395–407.

Reviewer 2 Report

Overall, this manuscript is well written and I have very few comments to make on it.

I would like the authors to make it more clear, who did the assessments for each of the patients. Was it differing specialists, and differing teams? In this question, I am trying to gauge the reliability of the measurements that were taken (as there were so many, probably taken by many different people).

My main suggestion is that the authors proof-read the manuscript to correct minor typos/clumsy sentences. There was the occasional word choice that seemed another word may have been a better selection eg. page 7 line 244 BMI was “seized”. Perhaps measured? Or even gathered? depending on the circumstances. My other suggestion would be for the authors to right align their left hand table column. The centred column is very hard to read. I would also suggest that in Table 1, the authors either choose to put Number or n, and percent or % - not both.

Author Response

R2: Overall, this manuscript is well written and I have very few comments to make on it.

A: We thank this reviewer for this positive comment on our study.

R2: I would like the authors to make it more clear, who did the assessments for each of the patients. Was it differing specialists, and differing teams? In this question, I am trying to gauge the reliability of the measurements that were taken (as there were so many, probably taken by many different people).

A: We thank this reviewer for this important comment. The body mass index (BMI) has been collected for every patient included in this study prior to surgery by the anaesthesiologist on duty (that changed several times over the years), as this is a necessity for anaesthesiologic procedures. Retrospectively, we have then collected these data (BMI) from the anaesthesiologic protocols and included them in our analysis. For every other single data set that has been analysed within this study we used the data collected by our cancer registry that is part of our surgical department. The data collected by our cancer registry has been collected by one single person (S.M.) over the last 20 years.

R2: My main suggestion is that the authors proof-read the manuscript to correct minor typos/clumsy sentences. There was the occasional word choice that seemed another word may have been a better selection eg. page 7 line 244 BMI was “seized”. Perhaps measured? Or even gathered? depending on the circumstances. My other suggestion would be for the authors to right align their left hand table column. The centred column is very hard to read. I would also suggest that in Table 1, the authors either choose to put Number or n, and percent or % - not both.

A: We thank this reviewer for these comments. We have proof-read the manuscript again and have it checked by a native speaker for correction of typos and phrases. As well, we have now left-aligned the left table column as aligning the column to the right side did not improve the readability of the table. We hope this reviewer agrees with the new style of the tables. We have adapted table 1 to the reviewers suggestions.